# Role of mTORC1 Signaling in Regulating the Immune Function of Granulocytes in Teleost Fish

**DOI:** 10.3390/ijms241813745

**Published:** 2023-09-06

**Authors:** Jiafeng Cao, Weiguang Kong, Gaofeng Cheng, Zhen Xu

**Affiliations:** 1Department of Aquatic Animal Medicine, College of Fisheries, Huazhong Agricultural University, Wuhan 430070, China; caojiafeng@nbu.edu.cn (J.C.); chenggaofeng@ihb.ac.cn (G.C.); 2Key Laboratory of Breeding Biotechnology and Sustainable Aquaculture, Institute of Hydrobiology, Chinese Academy of Sciences, Wuhan 430072, China; kongweiguang@ihb.ac.cn

**Keywords:** granulocytes, mechanistic target of rapamycin complex 1 (mTORC1), immunity, RNA-Seq, largemouth bass

## Abstract

Granulocytes are crucial innate immune cells that have been extensively studied in teleost fish. Studies in mammals have revealed that mechanistic target of rapamycin complex 1 (mTORC1) signaling acts as a significant immune regulatory hub, influencing granulocyte immune function. To investigate whether mTORC1 signaling also regulates the immune function of granulocytes in teleost fish, we established a model of RAPA inhibition of the mTORC1 signaling pathway using granulocytes from largemouth bass (*Micropterus salmoides*). Our results demonstrated that inhibition of mTORC1 signaling promoted autophagy and apoptosis of granulocytes while inhibiting cell proliferation. Moreover, inhibition of the mTORC1 signaling pathway enhanced the phagocytosis capacity of granulocytes. Collectively, our findings revealed the evolutionarily conserved role of the mTORC1 signaling pathway in regulating granulocyte responses, thus providing novel insights into the function of granulocytes in teleost fish.

## 1. Introduction

Granulocytes are essential phagocytes found in various vertebrates, including fish, amphibians, reptiles, birds, and mammals. These cells play a fundamental role in the innate immune defense against various pathogenic microorganisms [1,2]. Whenever tissue damage, inflammation, or pathogen infection occurs, granulocytes are rapidly recruited to the site of inflammation, where they employ multiple complementary mechanisms to eliminate pathogens [3,4]. Upon activation at the site of inflammation, neutrophils initiate phagocytosis, release toxic intracellular granules [5,6], generate reactive oxygen species (ROS) [7,8,9], and deploy neutrophil extracellular traps (NETs) [10,11] to efficiently eradicate pathogens. These characteristics are conserved both in teleost fish and mammals [12,13,14,15,16,17]. As a vital component of the innate immune system, granulocytes are well known for their potent ability to clear infections [2,4,18]. In teleosts, studies have demonstrated the role of granulocytes in resisting pathogenic invasion, including phagocytosis of pathogenic bacteria [19], infected cells [20], and intracellular clearance, as well as the capture of pathogenic bacteria through NETs to inhibit replication and infection [21,22].

The mechanistic target of rapamycin (mTOR) is a highly conserved protein (~280 kDa) found in all eukaryotes [23]. mTOR forms two complexes known as mTOR complex 1 (mTORC1) and mTOR complex 2 (mTORC2) by associating with different ligands. Interestingly, recent studies have reported confirmation of the presence of mTORC3, a complex that exhibits bimodal mTORC1/2 activity but lacks key mTORC1/2 components [24]. The main component of mTORC3 signaling that has been reported so far is mEAK-7 [25,26,27], although ETV7 [24] and GIT1 [28] have also been reported to be involved in the assembly of mTORC3, which still needs to be further explored. mTORC1 regulates protein synthesis, cell growth, and immunity by phosphorylating substrate proteins S6K1 and 4EBP1 [29,30,31]. Notably, mTORC1 is sensitive to inhibition by rapamycin (RAPA). In contrast, mTORC2 regulates cellular survival and the cytoskeleton through the phosphorylation of AGC kinases such as AKT, SGK1, and PKC-α, and its inhibition requires prolonged RAPA exposure [30,31,32,33]. Additionally, loss of mTORC3 exhibited cellular sensitivity to RAPA [24]. Although most studies on mTORC1-mediated regulation of immune function have focused on T and B cells, its role in granulocytes requires further exploration [34,35,36,37]. Studies in mammals have indicated that mTOR has an important role in regulating cell proliferation and migration [25,26,27], and mTORC1 signaling is involved in granulocyte activation [38], chemotaxis [39], migration [40,41], and recruitment [35,42]. Moreover, research has demonstrated that inhibition of mTORC1 signaling can lead to increased neutrophil autophagic activity, which mediates the release of NETs [43,44]. Nevertheless, the molecular mechanisms that regulate granulocytes through mTORC1 signaling have only thus far been identified in mammals. Given the evolutionary conservation of mTORC1 in all eukaryotes, understanding whether and how mTORC1 signaling regulates granulocytes in non-mammalian models is crucial. Considering the similarity in granulocyte function and the evolutionary conservation of mTORC1 signaling between teleosts and mammals, we hypothesized that similar molecular mechanisms regulating granulocyte immune responses and maintaining homeostasis through mTORC1 must have evolved in both modern and primitive bony vertebrates.

To investigate whether and how mTORC1 signaling regulates granulocyte immune responses, we utilized largemouth bass (*Micropterus salmoides*) as an early vertebrate model. Our results revealed that RAPA treatment effectively inhibited mTORC1 signaling, as evidenced by the downregulation of S6 and 4EBP1 phosphorylation in granulocytes. Furthermore, the RAPA treatment had a significant impact on several key signaling pathways in granulocytes, including mTOR, cell cycle, apoptosis, autophagy, endocytosis, and phagosome signaling pathways, as well as corresponding changes in cellular physiological phenomena. Notably, the mTORC1 signaling pathway also plays a critical role in granulocyte immune responses, particularly affecting phagocytosis. Our findings provide essential evidence elucidating the mechanism of mTORC1 regulation in granulocyte immune and physiological responses within teleost models. Therefore, this study contributes significantly to our understanding of the mTORC1 signaling pathway and its influence on granulocyte function from an evolutionary perspective, offering crucial insights into the functional evolution of granulocytes in early vertebrates.

## 2. Results

### 2.1. Granulocyte Isolation and In Vitro Inhibition of the mTORC1 Signaling Pathway in Granulocytes by RAPA

To investigate the involvement of the mTORC1 signaling pathway in largemouth bass granulocyte responses, we isolated head kidney granulocytes using FACS (Figure 1A). Flow cytometry analysis confirmed the high purity of the isolated granulocytes and non-granulocytes (Figure 1B–D). Subsequently, we examined the phosphorylation and protein levels of S6 and 4EBP1 in granulocytes after RAPA treatment through immunoblotting. As expected, we observed that while the protein levels of S6 and 4EBP1 remained unchanged in granulocytes of the RAPA-treated group compared to the DMSO-treated group, their phosphorylation levels were significantly inhibited (Figure 1E–G). These results strongly suggest that the mTORC1 signaling pathway in largemouth bass granulocytes can be inhibited by RAPA.

### 2.2. Analysis of Transcriptomic Changes in Largemouth Bass Granulocytes after In Vitro RAPA Treatment

To further analyze the effects of RAPA treatment on granulocytes, we sequenced RNA-Seq libraries using the Illumina platform. Through statistical analysis, we identified 3984 DEGs in the RAPA/DMSO group, comprising 1978 upregulated and 2006 downregulated genes. The expression pattern of these DEGs was then visualized using volcano plots (Figure 2A). A KEGG pathway enrichment analysis was conducted to gain a comprehensive understanding of the changes in DEGs following RAPA treatment of granulocytes. As illustrated in Figure 2B, many important signaling pathways were successfully enriched. The analysis revealed enrichment in signaling pathways associated with granulocyte immunity (e.g., endocytosis and phagosome (which is associated with the phagocytosis), regulation of actin cytoskeleton and lysosome (which is associated with the formation of NETs, etc.), granulocyte life activities (cell cycle, apoptosis, autophagy, necroptosis, ribosome biogenesis, DNA replication, RNA transport, etc.), as well as the classic signaling pathways (which control the growth of cells) affected after RAPA treatment (e.g., mTOR signaling pathway, MAPK signaling pathway, Wnt signaling pathway, etc.). To delve deeper into the effects of RAPA treatment on KEGG pathways, we next identified the enriched upregulated (Figure 2C) and downregulated (Figure 2E) signaling pathways, respectively. The upregulated pathways (such as endocytosis, regulation of the actin cytoskeleton (which is associated with the formation of NETs), autophagy, lysosomes, and necroptosis) after RAPA treatment suggested potential promotion of granulocyte phagocytosis and the formation of NETs influenced cellular life activities (acceleration of cellular senescence and autophagy) (Figure 2C). On the other hand, the downregulated pathways after RAPA treatment were indicative of potential inhibition of the cell cycle, ribosome biogenesis, DNA replication, and RNA transport, thereby hindering the progress of cellular life activities. (Figure 2E). In mammals, studies have also shown that changes in these pathways are closely related to the mTOR signaling pathway. To further emphasize the effect of RAPA treatment on granulocyte physiological and immune functions, 17 major upregulated (Figure 2D) and downregulated genes (Figure 2F) were selected from each of these significant pathways. Furthermore, the reliability of our transcriptome results was further confirmed by RAPA’s role as a targeted inhibitor of mTOR, which was consistent with the inhibition results of the mTOR signaling pathway and its heatmap (Appendix A). These findings provide valuable insights into the impact of RAPA treatment on largemouth bass granulocytes and their immune and physiological responses.

### 2.3. Inhibition of mTORC1 Signaling by RAPA Suppresses the Cell Cycle and Proliferative Capacity of Granulocytes

In mammals, the mTORC1 signaling pathway has been recognized for its crucial role in promoting cell growth and survival [31]. To determine whether mTORC1 serves a similar function in teleosts, we analyzed the transcriptome results and found that RAPA treatment of granulocytes inhibited the cell cycle signaling pathway (Figure 3A), with significant suppression of some important genes in this pathway (Figure 3B). Additionally, signaling pathways related to DNA replication and RNA transport were also inhibited after RAPA treatment (Figure 2E). Based on these transcriptome findings, we hypothesized that the inhibition of mTORC1 signaling by RAPA could suppress the cell cycle progression of granulocytes. To validate this hypothesis, flow cytometry analyses were conducted to assess cell proliferation. Our findings indicated that the proportion of proliferating granulocytes (relative to the total granulocytes) decreased from 1.4% to 0.8% in the RAPA-treated group (Figure 3C–E). These findings provide compelling evidence that inhibition of the mTORC1 signaling pathway effectively impedes cell cycle progression and cell proliferation in granulocytes.

### 2.4. Inhibition of mTORC1 Signaling by RAPA Induces Apoptosis and Autophagy in Granulocytes

The transcriptome results highlighted the inhibition of apoptosis and autophagy signaling pathways in granulocytes following RAPA treatment (Figure 4A,B and Appendix A), which has also been observed in mammals. This led us to speculate that mTORC1 signaling might regulate cell death in teleosts as well. To verify this hypothesis, we analyzed apoptosis in granulocytes using flow cytometry. While the proportion of early apoptosis granulocytes remained unchanged in the RAPA-treated group compared to the DMSO group (Figure 4C–E), the proportion of late and total apoptosis granulocytes increased by 3.68% and 3.29%, respectively, in the RAPA-treated group (Figure 4C–E). These results support the notion that inhibition of the mTORC1 signaling pathway induces cell apoptosis and autophagy in granulocytes.

### 2.5. Inhibition of the mTORC1 Signaling Pathway by RAPA-Promoted Phagocytosis in Granulocytes

The phagocytic function of granulocytes has been extensively studied in both mammals and teleosts. To investigate whether the mTORC1 signaling pathway is involved in the phagocytosis function of teleost granulocytes, we analyzed the transcriptome data and observed that the endocytosis and phagosome signaling pathways were highly enriched after RAPA treatment of granulocytes (Figure 5A,B and Appendix A). To further examine this hypothesis, we assessed the phagocytic ability of granulocytes using flow cytometry. The proportion of phagocytic granulocytes increased from 40.13% in the DMSO group to 44.40% after RAPA treatment in vitro (Figure 5C–E). Moreover, the inhibition of the mTORC1 signaling pathway significantly increased the percentage of high-capacity phagocytic granulocytes (ingestion of two or more beads per cell) from 46.34% to 51.82% (Figure 5C,D,F). These findings indicate that the mTORC1 signaling pathway plays a key role in regulating the number and intensity of phagocytic granulocytes, contributing to the immune response of teleost granulocytes.

## 3. Discussion

In vertebrates, granulocytes act as the first line of defense in innate immunity, and their core function involves being recruited to infection sites, recognizing and phagocytosing microorganisms, and subsequently eliminating pathogens through various cytotoxic mechanisms. These mechanisms include the generation of ROS, the release of AMPs, and the recently discovered process of expelling nuclear contents to form NETs [2,45]. In mammals, the mTORC1 signaling pathway has been implicated in the innate immune response of granulocytes [38,39,40,41,42,43,44]. However, little is known regarding whether mTORC1 signaling regulates granulocytes in teleost fish. Given the conservation of granulocyte function and mTORC1 signaling throughout vertebrate evolution, we hypothesized that mTORC1 signaling is evolutionarily conserved in regulating granulocyte immunity to maintain homeostasis. Therefore, our study sought to investigate whether and how the mTORC1 signaling pathway regulates the granulocyte immune response in teleost fish.

To explore the regulatory role of the mTORC1 signaling pathway in teleost granulocytes, we sorted largemouth bass granulocytes using FACS. As expected, similar to mammals, mTORC1 signaling in granulocytes was significantly inhibited after acute RAPA treatment, as evidenced by the reduced phosphorylation of S6 and 4EBP1. These results demonstrate that the mTORC1 signaling pathway in teleost granulocytes is affected by RAPA, suggesting that RAPA can be used as a specific inhibitor to investigate the effect of the mTORC1 signaling pathway on the function of teleost granulocytes. Additionally, it should be noted that long-term or high-dose RAPA treatment might alter mTORC2 assembly and activity. In our experiments, we confirmed that RAPA treatment (100 nM) for 30 min inhibited the mTORC1 signaling pathway in granulocytes, which is known to regulate granulocyte function. However, high-dose RAPA exposure did not have any appreciable effects on mTORC2. It has been reported that certain cell lines with RAPA-sensitive AKT/PKB phosphorylation (such as PC3, BJAB, and Jurkat) experience impaired mTORC2 integrity at 1 h after RAPA (100 nM) treatment and almost complete loss of the complex at 24 h [33]. Although we treated granulocytes with RAPA (100 nM) for 30 min, there is still the possibility that RAPA may affect the mTORC2 signaling pathway in granulocytes. However, compared to mammals, few studies have characterized the inhibitory effects of RAPA on the mTORC2 signaling pathway in teleost granulocytes. Therefore, additional research is needed to explore this intriguing hypothesis. Moreover, since acute RAPA treatment primarily affects the mTORC1 signaling pathway, we mainly focused on discussing the impact of the mTORC1 signaling pathway on granulocyte function. To assess the potential regulatory role of the mTORC1 signaling pathway in granulocytes, we analyzed the effects of RAPA-mediated mTORC1 inhibition on granulocytes at both the transcriptome and cellular levels.

Next, we proceeded to sequence the transcriptome of granulocytes treated with RAPA for 30 min in vitro. Excitingly, the mTORC1 signaling pathway, the target pathway of RAPA, was successfully enriched, confirming the reliability of the transcriptome results. Moreover, the pathways enriched after RAPA treatment were mainly related to essential cell functions such as cell cycle, apoptosis, autophagy, ribosome biogenesis in eukaryotes, ribosome, DNA replication, and RNA transport. This finding is noteworthy, as similar signaling pathways related to cell activities are generally regulated by the mTORC1 signaling pathway in mammals [31]. Therefore, this result provides substantial evidence that the involvement of the mTORC1 signaling pathway in regulating cell activities is evolutionarily conserved. Notably, pathways related to granulocyte phagocytosis were also enriched, including endocytosis, phagosome, lysosome, NOD-like receptor signaling pathway, and C-type lectin receptor signaling pathway. Although previous studies have reported the involvement of the mTORC1 signaling pathway in the phagocytosis of dendritic cells and macrophages [46,47], little is known regarding the regulation of granulocyte phagocytosis in vertebrates. In this study, we identified a potential association between the mTORC1 signaling pathway and granulocyte phagocytosis.

The coordination between cell growth and proliferation is vital to ensure normal cell cycle progression and survival, and conflicting signals may induce cell death or senescence [48,49]. mTOR, as a master regulator of cell metabolism, can respond to both intracellular and extracellular signals, playing a key role in coordinating cell growth and division [49]. Although the mechanisms through which the mTORC1 signaling pathway regulates cell activities in mammals have been extensively studied, little is known regarding its regulatory role in teleosts. Based on our transcriptome analysis results related to cell cycle, apoptosis, and autophagy signaling pathways, we next investigated the cell cycle, proliferation, apoptosis, and autophagy of granulocytes after RAPA treatment in vitro to further explore the effects of the mTORC1 signaling pathway on granulocyte functions in teleosts. Our data indicated that RAPA treatment of granulocytes could arrest the cell cycle process and inhibit cell proliferation. Previous studies have shown that the mTORC1 downstream effectors 4EBP1 and S6K1 regulate G1/S phase progression by mediating G1 cyclin transcription [50]. Activation of the mTORC1 signaling pathway in cells can upregulate c-Myc [51,52] and PCNA [53], thereby affecting the cell cycle process and promoting cell proliferation. Moreover, studies have demonstrated that RAPA inhibits epithelial cell proliferation by downregulating CDK1 and cyclins (cyclin A, cyclin B, and cyclin E), providing evidence that mTORC1 signaling plays a crucial role in cell proliferation [54]. In line with the results of downregulated genes in the cell cycle signaling pathway and the cell cycle and proliferation experiments, it appears that, similar to mammals, the mTORC1 signaling pathway in teleosts plays a significant role in regulating cell cycle progression and cell proliferation. Our results also demonstrated that inhibition of granulocyte mTORC1 signaling could promote apoptosis and induce autophagy. In mammals, mTORC1 signaling is involved in apoptosis and autophagy regulation [31,55,56]. Pro-apoptosis genes such as *bad, bax, caspase 8, and caspase 9* are upregulated during apoptosis [57]. Moreover, mTORC1 can inhibit autophagy by activating ULK1 to suppress the catabolic process [58], and ATG genes also play an important role in regulating autophagy [59]. Therefore, based on our experimental results, it appears that the mTORC1 signaling pathway is closely associated with apoptosis and autophagy in both mammals and teleosts. Although our study only provided a brief exploration at the transcriptome and cellular level, the above-described data collectively suggest that mTORC1 signaling plays an evolutionarily conserved role in regulating granulocyte activities in vertebrates.

Phagocytosis by phagocytes is a crucial defense mechanism of the body, enabling the devouring and destruction of particulate matter [60]. In mammals, the regulatory role of mTORC1 signaling in the phagocytosis of dendritic cells and macrophages has been extensively studied [46,47]. However, despite extensive research on granulocyte phagocytosis in recent decades, it remains unclear whether mTORC1 signaling also regulates the phagocytosis of granulocytes. Our results demonstrated a significant increase in the percentage of phagocytic granulocytes after RAPA treatment. Interestingly, the phagocytic capacity, measured as the average number of beads internalized by phagocytic granulocytes, was also significantly enhanced after inhibiting mTORC1 signaling. Therefore, our findings provide the first evidence that inhibiting mTORC1 signaling can substantially increase the phagocytosis of teleost granulocytes. Similar results have been observed in mammals, where RAPA-induced inhibition of mTORC1 significantly enhanced the phagocytic ability of microglia [61]. However, it should be noted that previous studies have reported conflicting findings. For example, a previous study reported that short-term exposure to RAPA may inhibit mTORC1 in mouse macrophages [47,62] and trout B cells [37], leading to a reduction in phagocytosis. These discrepancies are likely due to differences in phagocytosis mechanisms among different phagocytes, suggesting that mTORC1 signaling may regulate phagocytosis through distinct pathways. Thus, further research is needed to investigate the specific regulatory mechanisms through which mTORC1 signaling controls the phagocytosis of different phagocytes.

## 4. Materials and Methods

### 4.1. Experimental Animals

Healthy largemouth bass (*Micropterus salmoides*) were obtained from a fish farm in E Zhou (E Zhou, China) and then maintained in a recirculating water system at Huazhong Agricultural University for over 4 weeks before the experiment. The fish (about 120 fish) were kept at 26 °C and fed daily with a commercial diet. Feeding was terminated 48 h prior to the sampling experiment.

### 4.2. Granulocyte Isolation

To gain granulocytes, we first isolated leukocytes from the head kidney as described previously [37] with modifications. Briefly, largemouth bass head kidneys were treated with DMEM (Gibco™, Gaithersburg, MD, USA) supplemented with 5% FBS (Gibco™, Amarillo, TX, USA) and mechanically disaggregated on a 100 μm nylon cell strainer (SPL life sciences, Gyeonggi-do, Republic of Korea). Then, the obtained cell suspensions were placed into a 34% to 51% percoll (Cytiva, Uppsala, Sweden) discontinuous density gradient and centrifuged at 400 g for 20 min at 4 °C. The kidney leukocytes at the interface were collected and washed with DMEM. After that, fluorescence-activated cell sorting (FACS) was adopted to sort granulocytes from head kidney leukocytes. Briefly, we performed flow sorting by selecting cells in the target region with FSC and SSC. The collected sorted granulocytes were backtested to verify the sorting purity and suspended in DMEM (supplemented with 5% FBS) pending further analysis.

### 4.3. Rapamycin (RAPA) Treatment

The largemouth bass (~50 g) were used for in vitro RAPA (Sigma-Aldrich, Louis, MO, USA) treatment studies. Head kidney granulocytes were suspended in DMEM (supplemented with 5% FBS) and divided into 48-well plates with 200 μL per well and a final density of 2 × 10^6^ cells/mL. RAPA was dissolved in DMSO (Sangon Biotech, Shanghai, China) at 10 mg/mL and diluted with PBS before use. The final applied concentration of RAPA in the RAPA group was 100 nM, and as a control, the same volume of DMSO was diluted with PBS in the DMSO group. Subsequently, the 48-well plates were placed in a cell incubator at 26 °C with 5% CO_2_ for 30 min, and the granulocytes were collected after centrifugation at 1000 g for the next analysis.

### 4.4. Sequencing and Analyses

The granulocyte samples of the DMSO group and the RAPA group were sent to Biomarker Technologies Co., Ltd. (Beijing, China). Briefly, the RNA was extracted using a TRIzol reagent (Invitrogen, Carlsbad, CA, USA), and the concentration and integrity of the RNA were checked by NanoDrop 2000 (Thermo Fisher Scientific, Wilmington, DE, USA) and the RNA Nano 6000 Assay Kit of the Agilent Bioanalyzer 2100 system (Agilent Technologies, CA, USA), respectively. Thereafter, the RNA was used for stranded RNA sequencing library preparation using a VAHTS Universal V6 RNA-seq Library Prep Kit for Illumina^®^ following the manufacturer’s instructions. PCR products were enriched, quantified, and finally sequenced on the illumina novaseq6000. The clean reads were mapped to the largemouth bass (*Micropterus salmoides*) genome. By using Hisat2’s soft tools, only reads with a perfect match or one mismatch were further analyzed and annotated based on the largemouth bass genome. Differential expression analysis of two groups was performed using DESeq2. The FDR ≤ 0.05 and Fold Change ≥ 2 were set as the thresholds for significantly differential expression. For further analysis of the DEGs, we carried out a KEGG enrichment using KOBAS software [63] to identify the major pathways that were significantly enriched after RAPA treatment.

### 4.5. Western Blot

To detect the inhibition of the mTORC1 signaling pathway in neutrophils, Western blots were performed. The intracellular proteins were resolved on a 12% SDS-PAGE gel (Bio-Rad, Hercules, CA, USA) under reducing conditions. Then the gels were transferred onto a PVDF membrane and blocked in PBS with 8% skim milk (Bio-Rad, Hercules, CA, USA). After that, the membrane was incubated with anti-S6 (2217), anti-4EBP1 (9644), anti-phospho-S6 Ser^240/244^ (5364), anti-phospho-4EBP1 Thr^37/46^ (2855) monoclonal rabbit antibody (Cell Signaling Technology company, Boston, MA, USA) or anti-β-actin (AC026) monoclonal rabbit antibody (Abclonal Technology, Wuhan, China) for 1 h on a shaker. After washing four times with PBST (5 min/time), goat anti-rabbit IgG was added and incubated for 45 min at room temperature. After washing four times with PBST (5 min/time), the reaction bands were visualized with the ECL substrate (Bio-Rad Laboratories) and then scanned by the Amersham Imager 600 Imaging System (GE Healthcare, Chicago, IL, USA). The densitometry of the target band was analyzed with ImageQuant TL software (GE Healthcare, Chicago, IL, USA).

### 4.6. Proliferation Assay

The proliferation assay was performed with Beyoclick^TM^ EDU-647 kit (Beyotime, Nantong, China) according to the manufacturer’s instructions. Briefly, to detect the percentage of proliferation granulocytes after RAPA treatment, the head kidney granulocytes were collected after 30 min of RAPA treatment, and 20 μg of 5-ethynyl-2′-deoxyuridine (EdU) (Invitrogen, Carlsbad, CA, USA) was added to the granulocyte culture media in vitro. After 1 d, the cells were fixed, permeabilized, and stained for EDU. Then, cells were analyzed using a CytoFLEX LX flow cytometer (Beckman Coulter, Brea, CA, USA).

### 4.7. Apoptosis Assay

The apoptosis assay was performed with the AnnexinV-FITC kit (Beyotime, Nantong, China) according to the manufacturer’s instructions. Briefly, to detect the effect of RAPA treatment on granulocyte apoptosis, the head kidney granulocytes were collected after 30 min of 100 nM DMSO or RAPA treatment. After washing two times, Annexin-V-FITC was added to the granulocytes, incubated for 15 min at room temperature away from light. After washing two times, PI (Beyotime, Nantong, China) was added to the granulocytes, incubated for 15 min at 4 °C, and analyzed using a CytoFLEX LX flow cytometer (Beckman Coulter, Brea, CA, USA).

### 4.8. Phagocytosis Assay

To detect the effect of RAPA treatment on granulocyte phagocytosis. The phagocytosis assay was performed as described previously with sight modifications [64]. Briefly, the head kidney granulocytes were collected after 30 min of 100 nM DMSO or RAPA treatment, and the granulocytes were co-incubated with 1.0 μm fluorescent beads (PolySciences, Palo Alto, CA, USA) with a cell/bead ratio of 1:20 in a cell incubator with 5% CO_2_ at 26 °C for 15 min. Cytochalasin D (Thermo Fisher Scientific, Wilmington, DE, USA) was added immediately to a final concentration of 5 μM, and the granulocytes were collected by centrifugation, discarding the supernatant. Then, 0.4% Taipan blue was added to quench the fluorescence of the free beads, incubated for 1 min, and then centrifuged to collect the granulocytes. The granulocytes were resuspended and immediately analyzed using a CytoFLEX LX flow cytometer (Beckman Coulter, Brea, CA, USA).

### 4.9. Statistical Analysis

Statistical analyses were conducted by GraphPad Prism 8 software. Data comparisons between two groups were detected by a paired Student’s *t*-test and one-way ANOVA. Data were expressed as the mean ± SEM, and *p <* 0.05 was considered statistically significant.

## 5. Conclusions

In conclusion, our study provides compelling evidence for a conserved role of mTORC1 signaling in the function of granulocytes in teleosts. Notably, inhibiting mTORC1 signaling in largemouth bass granulocytes significantly affected cell cycle progression, proliferation, apoptosis, and autophagy. Moreover, our results demonstrate for the first time that mTORC1 signaling plays a crucial regulatory role in the phagocytosis of teleost granulocytes. Given the functional analogy between granulocytes in fish and mammals, our findings suggest that the elucidated mTORC1 signaling pathway plays an evolutionarily conserved role in regulating granulocyte responses, thus providing valuable insights into the function of granulocytes in teleost fish.

## Figures and Tables

**Figure 1 ijms-24-13745-f001:**
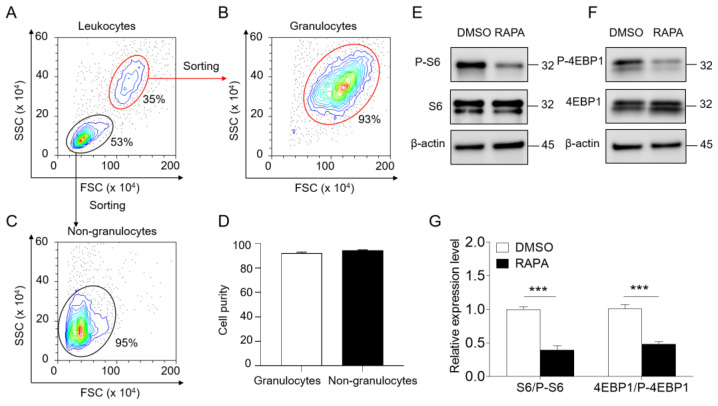
RAPA can inhibit mTORC1 signaling in largemouth bass neutrophils in vitro. (**A**–**C**) Sorting of granulocytes and non-granulocytes by FACS. (**D**) Ratio of granulocyte and non-granulocyte sorting (*n* = 6 fish/group). (**E**,**F**) Immunoblotting analysis showing total protein or phosphorylation levels of the indicated mTORC1 components containing S6 (**E**) and 4EBP1 (**F**) in granulocytes. β-actin was used as a loading control. (**G**) The relative ratio of phosphorylation levels to those of total protein levels of S6 and 4EBP1 in granulocytes, evaluated by densitometric analysis of immunoblots from representative e and f, respectively (*n* = 11 fish/group). Statistical differences were analyzed by a paired Student’s *t*-test. Data in (**G**) are representative of at least three independent experiments (mean ± SEM). *** *p* < 0.001.

**Figure 2 ijms-24-13745-f002:**
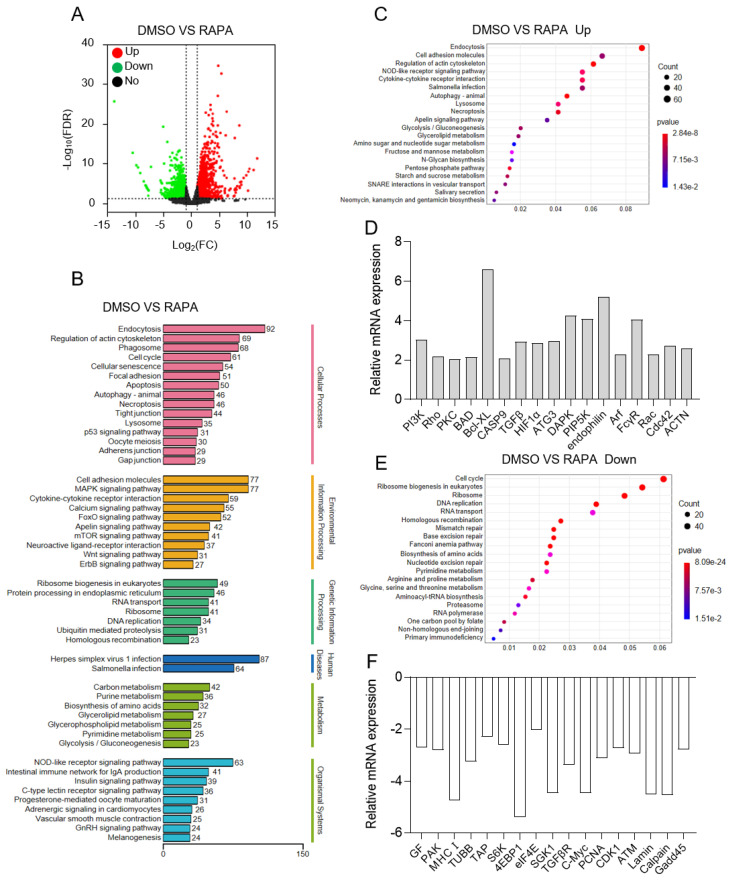
Effect of RAPA treatment on transcriptomic changes in head kidney granulocytes of largemouth bass. (**A**) Volcano plot displaying the DEG distribution in the RAPA treatment group compared with the control group. (**B**) The number of DEGs in each KEGG pathway. Different colors represent different categories. (**C**,**E**) The major upregulated (**C**) and downregulated (**E**) KEGG pathways in the RAPA treatment group compared with the control group. (**D**,**F**) The expression of differential genes in the KEGG pathway that were mainly upregulated (**D**) and downregulated (**F**) in the RAPA treatment group compared with the control group. Positive numbers on the Y axis mean upregulation, while negative values mean downregulation.

**Figure 3 ijms-24-13745-f003:**
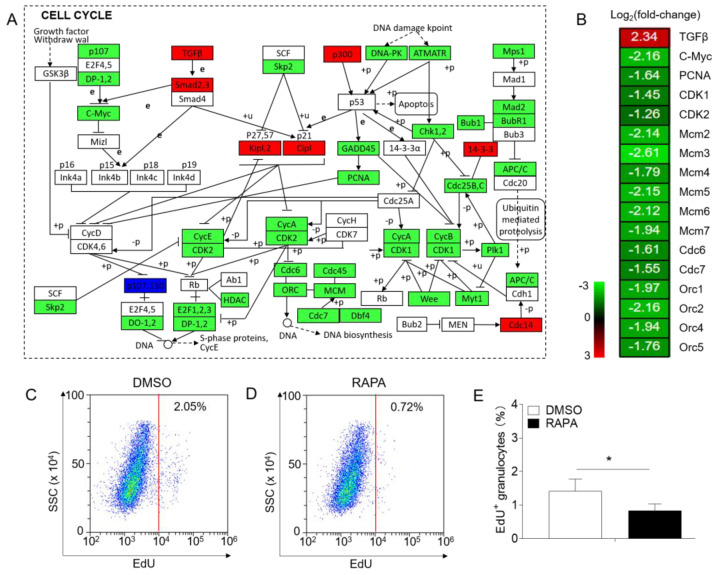
Effects of the in vitro treatment of granulocytes with RAPA on cell cycle signaling pathways and cell growth. (**A**) Both red, green, and deep blue shading boxes represent molecules of the cell cycle pathway identified in head kidney neutrophils of largemouth bass. Additionally, the red boxes indicate the upregulated DEGs in this pathway, the green boxes indicate the downregulated DEGs in this pathway, and the deep blue boxes indicate both upregulated and downregulated DEGs in this pathway. (**B**) Differential expression genes involved in the cell cycle pathway were analyzed after RAPA treatment. The color gradient represents highly upregulated (red) and highly downregulated (green) genes. (**C**,**D**) Representative flow cytometry dot plot showing proliferation of granulocytes treated with DMSO (**C**) or RAPA (**D**). (**E**) The ratio of EdU+ granulocytes from granulocyte populations treated with DMSO or RAPA (*n* = 6 fish/group). Statistical differences were determined by a paired Student’s *t*-test. Data in (**E**) are representative of at least three independent experiments (mean ± SEM). * *p* < 0.05.

**Figure 4 ijms-24-13745-f004:**
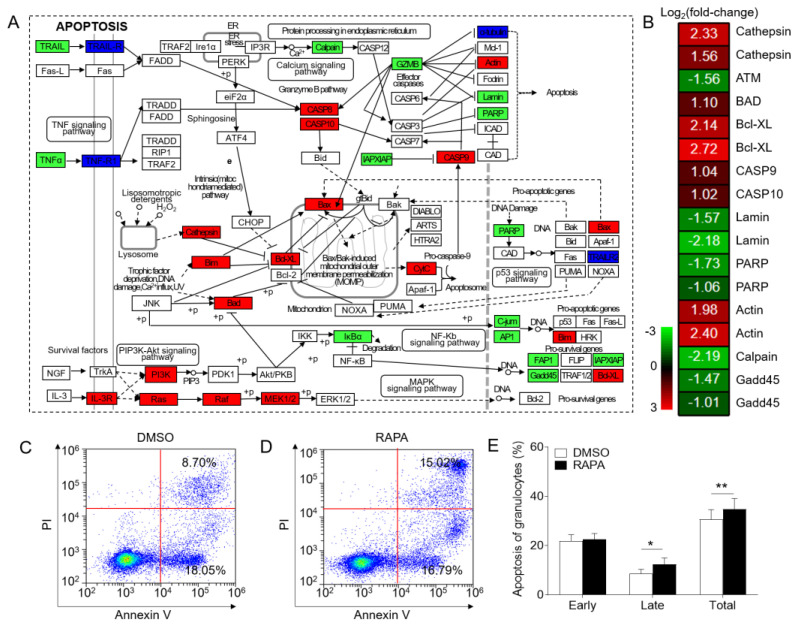
Effect of in vitro treatment of granulocytes with RAPA on apoptotic signaling pathways and cell apoptosis. (**A**) Both red, green, and deep blue shading boxes represent molecules of the apoptosis pathway identified in head kidney granulocytes of largemouth bass. Additionally, the red boxes indicate the upregulated DEGs in this pathway, the green boxes indicate the downregulated DEGs in this pathway, and the deep blue boxes indicate both upregulated and downregulated DEGs in this pathway. (**B**) Differential expression genes involved in the apoptosis pathway were analyzed after RAPA treatment. The color gradient represents highly upregulated (red) and highly downregulated (green) genes. (**C**,**D**) Representative flow cytometry dot plot showing apoptosis of granulocytes treated with DMSO (**C**) or RAPA (**D**). (**E**) The ratio of apoptosis granulocytes from granulocyte populations treated with DMSO or RAPA (*n* = 6 fish/group). Statistical differences were determined by a paired Student’s *t*-test. Data in (**E**) are representative of at least three independent experiments (mean ± SEM). * *p* < 0.05, ** *p* < 0.01.

**Figure 5 ijms-24-13745-f005:**
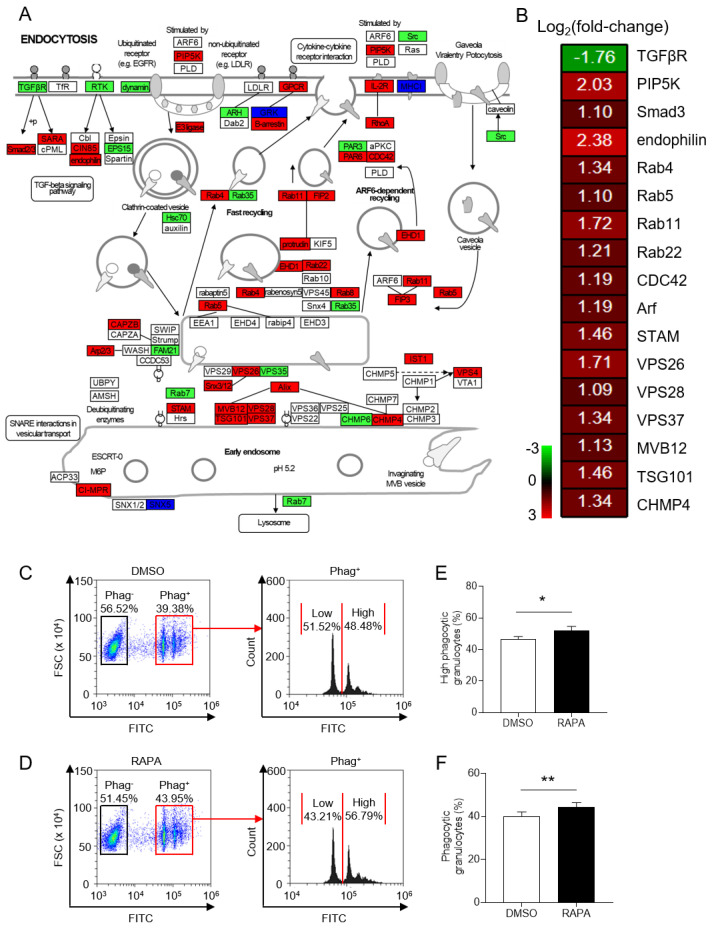
Effect of in vitro treatment of granulocytes with RAPA on phagocytosis signaling pathways and cell phagocytosis. (**A**) Both red, green, and deep blue shading boxes represent molecules of the phagocytosis pathway identified in head kidney granulocytes of largemouth bass. Additionally, the red boxes indicate the upregulated DEGs in this pathway, the green boxes indicate the downregulated DEGs in this pathway, and the deep blue boxes indicate both upregulated and downregulated DEGs in this pathway. (**B**) Differential expression genes involved in the phagocytosis pathway were analyzed after RAPA treatment. The color gradient represents highly upregulated (red) and highly downregulated (green) genes. (**C**,**D**) Representative flow cytometry dot plot showing phagocytosis of granulocytes treated with DMSO (**C**) or RAPA (**D**). Phag−, nonphagocytic; Phag+, phagocytic. “Low” indicates phagocytic granulocytes internalizing one bead, and “High” represents phagocytic granulocytes internalizing two or more beads. (**E**) The ratio of phagocytic granulocytes from granulocyte populations treated with DMSO or RAPA (*n* = 6 fish/group). (**F**) The ratio of high-phagocytic granulocytes internalizing two or more beads from phagocytic granulocyte populations treated with DMSO or RAPA (*n* = 6 fish/group). Statistical differences were determined by a paired Student’s *t*-test. Data in (**E**,**F**) are representative of at least three independent experiments (mean ± SEM). * *p* < 0.05, ** *p* < 0.01.

## Data Availability

All original data can be requested from the first author.

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
