# Peer review of "Role of mTORC1 Signaling in Regulating the Immune Function of Granulocytes in Teleost Fish"

_ijms, 2023, doi:10.3390/ijms241813745_

Round 1
Reviewer 1 Report
The authors, Cao et al, have prepared an interesting article on the “Role of mTORC1 signaling in regulating immune function of granulocytes in teleost fish”. By treating granulocytes from teleost with DMSO and Rapamycin, an mTORC1 inhibitor, they demonstrate an essential role for immune cell regulation. Rapamycin is a potent inhibitor of the immune cell function in mammals, as demonstrated through the medical breakthrough use of rapamycin in transplant medicine. They show that rapamycin inhibits cell cycle and growth through flow cytometry, and also provide evidence of apoptosis. While this body of work is thorough, more experiments are required to improve the impact of the findings.
Revisions:
1) In the second paragraph of the introduction, please summarize and cite the following manuscripts (PMID: 29750193, 31288154, 33364499). Introducing mTORC3 signaling will be important background information.
2) Figure 1 E and F: Can authors provide the phosphorylation site of p-S6 and p-4E-BP1? mTOR specific phosphor-site for S6 is Serine 240/244 and 4E-BP1 is Threonine 37/46. Also, showing p-S6K1 Threonine 389 and p-Akt Serine 473 will be essential as well. There is a meak-7 antibody that exists (Origene Technologies: #TA501037) for human, but I don’t know if it works for teleost. It would be worth checking though.
3) Figure 3. Please provide a direct cell count or active measurement of cell proliferation with DMSO or Rapa in the granulocytes.
4) Figure 4. Please provide a western blot showcasing cleaved caspase 3 for apoptosis, or other equivalent markers.
5) Please provide antibodies and catalog numbers for all targets.
6) Please provide the raw western blot data in the supplemental figures, annotated.
Acceptable
Author Response
We would like to thank the editors and reviewers for their insightful and constructive comments. Please see the attachment, thanks!

Reviewer 2 Report
In the article «Role of mTORC1 signaling in regulating immune function of granulocytes in teleost fish», the authors presented interesting and original data on largemouth bass with an attempt to compare the involvement of mTORC1 signaling with mammalian data. Despite the complex design of the presented work, the review process revealed numerous shortcomings that do not allow us to recommend this work in its current form for publication in IJMS. The authors of the work should also take into account that copy protection of drawings makes it difficult to review the article.
1. In fig. 1 it is not clear the ratio of granulocytes and agranulocytes why their values for each subtype are about 100%. What, in this case, was taken as a starting point?
2. Why for the data on immunoblotting in Fig. 1E and F do not indicate the molecular weights of the studied proteins?
3. In fig. 2 presents extensive information that is poorly analyzed. It is necessary to give a more comprehensive description of section 2B and justify the expediency of its use in this work.
4. The main upstream (2C) and downstream (2E) KEGG pathways in the RAPA treatment group should be carefully analyzed compared to the control group. Just a brief mention of these data in the "Results" section is not enough to understand the need to use these parameters in the work.
5. The same applies to data on the expression of differential genes in the KEGG pathway, which were mainly upregulated (2D) and downregulated (2F) in the RAPA treatment group compared to the control group. It is necessary to give a clear justification why these particular genes were chosen and how they were further analyzed.
6. Fig. 3A is not readable, you need to reformat it and increase the legibility and font size of the lettering. Why do the blue rectangle in fig. 3, the authors described as plural?
7. Fig. 4A is not readable, you need to reformat it and increase the legibility and font sizes of the lettering.
8. In section 2.4. the authors do not provide convincing evidence for their hypothesis. Considering that Fig. 4A is illegible, it is difficult to compare the data on mammals declared by the authors and sea bass. This section is not conclusive.
9. Fig. 5A is not readable, you need to reformat it and increase the legibility and font size of the lettering. Captions for fig. 5A does not present the dates of presentation of the content of this fragment of the figure.
10. How many fish were used in this work?
11. It is necessary to indicate the catalog numbers of all reagents used in the work.
12. Contents of Sections 4.5. Western blot and 4.6. Proliferation assay, does not live up to their names. It is necessary to provide adequate information in these sections.
13. Sections 4.7. Apoptosis analysis and 4.8. The analysis of phagocytosis, also needs to be carefully checked and a complete characterization of these methodological steps.
The quality of the English language needs improvement
Author Response

(The authors gave the same response as above.)

Round 2
Reviewer 1 Report
Hello, please fix the introduction.
mEAK-7 is an essential component of mTORC3 signaling. There are several articles that discuss an additional mTOR complex, but ETV7, GIT1, etc, have not been further validated. You may mention these, but please modify your introduction.
Reviewer 2 Report
The corrected version of article may be published in IJMS.
